published: 20 May 2021

# Sunburn, Sun Safety and Indoor Tanning Among Schoolchildren in Ireland

András Költő[1]*, Lauren Rodriguez[2], Helen McAvoy[2] and Saoirse Nic Gabhainn[1]

[1]Health Promotion Research Centre, National University of Ireland Galway, Galway, Ireland, [2]Institute of Public Health, Dublin, Ireland

**Objectives:** We present patterns of sunburn, sun safety behaviors and indoor tanning bed use in a nationally representative sample of schoolchildren aged 10–17. These behaviors were explored across gender, age, and social class groups.

**Methods:** Within the Health Behaviour in School-aged Children (HBSC) Ireland study, 10,271 young people (aged 13.54 ± 1.92, percentage girls 53.3%) reported frequency of sun safety behaviors, sunburn, and frequency and circumstances of indoor tanning bed use.

**Results:** Children frequently experienced sunburn (90% lifetime, 74% last year), and 3% reported never using any sun protection. Applying sunscreen and wearing sunglasses were the most commonly used sun safety measures; other ways of sun protection were less popular. Indoor tanning bed use was reported by around 5%, and a large proportion of users were not advised of any indoor tanning safety measures. Sun safety behaviors varied by age and gender, with some socio-economic differences in tanning bed use. An association was found between frequency of family holidays abroad and sunburn.

**Conclusion:** Targeted interventions are needed to increase sun safety behaviors and eliminate tanning bed use among children in Ireland.

Keywords: adolescents, HBSC, sunburn, sun safety, indoor tanning beds, UV exposure, sunscreen use, skin cancer

**Edited by:**
Sonja Merten,
Swiss Tropical and Public Health
Institute, Switzerland

**\*Correspondence:**
András Költő
andras.kolto@nuigalway.ie

**Citation:**
Költő A, Rodriguez L, McAvoy H and
Nic Gabhainn S (2021) Sunburn, Sun
Safety and Indoor Tanning Among
Schoolchildren in Ireland.
Int J Public Health 66:1604045.

## INTRODUCTION

Excessive exposure to ultraviolet (UV) radiation, sunburn and indoor tanning bed use in childhood are well established risk factors for developing skin cancer later in life [1]. Many skin cancers are preventable through a combination of sun safety behaviors including wearing sunscreen, avoiding peak UV hours of the day, wearing protective clothing which cover arms and legs, and wearing a hat [2]. Evidence suggests that employing a combination of these behaviors during the critical period of childhood and adolescence can reduce the risk of subsequent skin cancer diagnosis later in life [3].

Over the past four decades, the incidence of skin cancer has increased throughout the world, with the highest rates found in Australasia, North America and Europe [4]. In Ireland, skin cancer incidence rates exceed rates for prostate and breast cancer, making skin cancer the most frequently diagnosed cancer on the island. Ireland ranks well above the EU average for rates of both malignant

melanoma incidence and mortality[1], and rates are projected to double by 2045 [3]. However, there is a lack of representative data on the prevalence of behaviors related to sun safety in children and adolescents in the Republic of Ireland. This presents an evidence gap for targeting effective public health responses and monitoring progress of skin cancer prevention in line with government policy [5]. In Ireland, the *Public Health (Sunbeds) Act 2014* prohibits the supply of commercial indoor tanning to people under 18, but prevalence of young people using indoor tanning devices still needs to be explored.

Experiences of sunburn, adoption of sun safety measures and indoor tanning bed use among young people have been investigated in Europe and the United States. Overall, sun safety behaviors are more likely to be adopted by girls than boys [6–8]. An English study found that age was associated with spending time outdoors. Younger children reported they had spent more time in the sun than older adolescents, however no gender differences were observed [9]. However, older children and adolescent girls have emerged as a higher risk group for use of indoor tanning [10, 11]. While age and gender differences are observed in sun safety behaviors, overall incidence of sunburn remain high. In Northern Ireland, 78% of children aged 11–16 years reported having a sunburn on at least one occasion in the previous year [12], and 19% reported three or more sunburn episodes. Studies in countries with an overall higher UV index, such as Australia and New Zealand, report similar sun safety behavior patterns [13, 14]. Boys were more likely than girls to stay outdoors without sunscreen, while girls were more likely than boys to use indoor tanning beds [15]. A pattern of decline in sun safety behaviors with increasing age has also been observed [13]. Social class is also associated with tanning risk behaviors. A Danish study found that risky tanning behavior was associated with high income, while indoor tanning bed use was observed more frequently among lower socio-economic groups [7]. The association of high income with higher UV exposure may be mediated by children from more affluent families spending family holidays abroad in countries with sunny climates, exposing children to intermittent high sun exposure [16].

This study presents patterns of sunburn, sun safety behaviors and indoor tanning use in a nationally representative sample of schoolchildren in Ireland and provides the first nationally representative evidence on UV-related behaviors in this age group. We collected information on behaviors such as sunscreen use, hat use, sunglasses use, the use of protective clothes and avoiding the sun by children aged 10–17. Baseline data on sunburn and indoor tanning bed prevalence are also reported. Two additional aims were to investigate whether socio-demographic factors are associated with UV-related behaviors, and whether having had family holiday(s) abroad is associated with the prevalence of sunburn.

---

[1]Estimates of cancer incidence and mortality in 2018 were obtained from the website of the European Commission Information Center: https://ecis.jrc.ec. europa.eu/explorer.php (Date of access: 19 January 2021).

# METHODS

## Procedure
Data were collected within the 2018 Irish Health Behaviour in School-aged Children (HBSC), a World Health Organization collaborative cross-cultural study that aims to map health behaviors and its psychosocial determinants in adolescents [17]. The survey was conducted in adherence to the protocol of the international HBSC network [18] in school classrooms, as described by Költő et al. [19]. The protocol stipulates new questions/items in the nationally representative surveys must be piloted in a smaller sample. Results of piloting the items on UV-related behaviors, conducted in Ireland in March–April 2018 with 363 children, are reported elsewhere [20]. Nationally representative data collection took place between April and September 2018.

For this study, questionnaires were administered in randomly selected school classrooms by teachers. Participating students and their parents/guardians and school principals gave informed consent prior to data collection. All children were informed that participation was voluntary, they were not obliged to answer any questions, and that all answers would be considered confidential. Children placed completed questionnaires into blank envelopes before collection and return to the research team. Ethical approval was received from the Research Ethics Committee of the National University of Ireland Galway.

## Sample
Irish schools are categorized into primary, which children attend for eight years, starting at the age of 4 or 5; and post-primary, which most children attend for a further six years. We analyzed data from children in the two most senior years in primary schools and in years 1–5 of post-primary schools who provided information on their age, gender, and socio-economic status. In total, 10,271 young people aged 10–17 were included. The mean age was 13.54 years (SD = 1.92); 53.3% were girls and 46.7% boys. Respondents were classified into three age groups: 10 to 11 year-olds (17.5%), 12 to 14 year-olds (49.1%), and 15 to 17 year-olds (33.4%). Based on parental occupation, 54.0% of the children were categorized as Social Class 1–2 (highest), 35.3% as Social Class 3–4 (medium) and 10.7% as Social Class 5–6 (lowest).

## Measures
### Sociodemographic Variables and Family Holidays Abroad
Children were asked to provide their gender (whether they are "a boy" or "a girl") and the year and month they were born. Following the definition of the Central Statistics Office [21], children of professional workers, and managerial and technical workers were classified into Social Class 1–2 (SC1-2); children of non-manual and skilled manual workers into Social Class 3–4 (SC3-4); and children of semi-skilled and unskilled workers into Social Class 5–6 (SC5-6). As part of the Family Affluence Scale, a six-item composite measure assessing family wealth developed by the international HBSC network [22], young people were asked

---

**1. If you go outside on a sunny day, do you...**

|  | Always | Sometimes | Never |
|---|---|---|---|
| use a hat? | ☐ | ☐ | ☐ |
| wear sunglasses? | ☐ | ☐ | ☐ |
| wear clothes that cover arms and legs? | ☐ | ☐ | ☐ |
| avoid the sun between 12 and 3pm? | ☐ | ☐ | ☐ |
| use sunscreen? | ☐ | ☐ | ☐ |

**2. How many times did you get sunburn (red skin for hours after being in the sun)?**

|  | Never | 1 time | 2 times | 3-4 times | 5 times or more |
|---|---|---|---|---|---|
| Last summer | ☐ | ☐ | ☐ | ☐ | ☐ |
| In your lifetime | ☐ | ☐ | ☐ | ☐ | ☐ |

**3. How many times have you used an indoor tanning bed (lying down or standing up)?**

|  | Never | 1 time | 2 times | 3-4 times | 5 times or more |
|---|---|---|---|---|---|
| In the last 12 months | ☐ | ☐ | ☐ | ☐ | ☐ |
| In your lifetime | ☐ | ☐ | ☐ | ☐ | ☐ |

**4. If you have used an indoor tanning bed (lying down or standing up), were you...**

|  | Never used a tanning bed | Yes, every time | Yes, at least once | No | I don't remember |
|---|---|---|---|---|---|
| asked how old you were? | ☐ | ☐ | ☐ | ☐ | ☐ |
| told to wear protective eye goggles? | ☐ | ☐ | ☐ | ☐ | ☐ |
| given advice on your skin type? | ☐ | ☐ | ☐ | ☐ | ☐ |
| told about the health risks? | ☐ | ☐ | ☐ | ☐ | ☐ |

**FIGURE 1 |** Sun- and UV-related survey items in the 2018 Health Behaviour in School-aged Children study in Ireland.

how many times they and their family had travelled out of Ireland for a holiday last year, with the response options "Not at all/Once/Twice/More than twice".

## Sun Safety

Since the sun- and UV-related items are novel to the HBSC Ireland study, we present the wording and the layout of these items in **Figure 1**. The first group of items asked about the frequency of using a hat, wearing sunglasses, wearing clothes that cover the limbs and avoiding sun during the peak UV hours (between noon and 3pm), and using sunscreen when going out on a sunny day, with the response options "Always/Sometimes/Never". In the development of the items, we considered measures used in other youth population health surveys. A similarly worded question – containing a different set of items and a five-point scale – was used in the Young Persons' Behaviour and Attitudes Survey (YPBAS) 2019 questionnaire [12]. The validated sun exposure questionnaire from Køster et al. [23] also contains a similar question (without the sunglasses item), but

it refers to vacations in a Southern European destination (e.g., Spain), and uses a five-point scale to assess agreement.

## Sunburn

Children were asked about the frequency of sunburn episodes (sunburn defined as "red skin for hours after being in the sun") last summer and in their lifetime, with response options being "Never/one time/two times/three-four times/five times or more" (**Figure 1**). A similar question was employed in the Growing Up Today Study, conducted in the United States in 1999 [24], although it asked only about sunburn episodes last summer, defined sunburn as "exposed parts of your skin stay red for several hours after you had been out in the sun", and the highest response option was three times or more. The Danish Sun Surveys 2009 and 2010 [25] used an analogous question for parents: "How many times has your child been sunburnt this summer?", with the same response options as used in our survey.

## Indoor Tanning Bed Use

The indoor device that emits UV light to tan the skin of its user has many different names: "solarium" [26], "indoor tanner" [27], "indoor tanning bed" [28], or "sunbed" [29]. Due to the lack of earlier research on indoor tanning bed use in youth in Ireland, we were unsure which term would be the most appropriate. The question wording was based on the feedback received during the pilot study: "How many times have you used an indoor tanning bed (lying down or standing up) in the last 12 months / in your lifetime?". The response options were same as those for sunburn episodes (**Figure 1**).

## Circumstances of Using an Indoor Tanning Bed

Based on earlier findings in the literature and in line with our research questions, we aimed to explore whether children had been asked about their age, told to wear protective goggles [12], received advice on their skin type [29], and instructed about health risks [29] when using an indoor tanning bed. For each item, the response options were: "Never used a tanning bed/Yes, every time/Yes, at least once/No/I don't remember" (**Figure 1**).

## Statistical Analysis

Analyses were carried out in SPSS 24.0. Overall descriptive analyses were conducted on the sun protection and tanning bed use variables. The associations of these variables with gender, age group and social class were examined by Chi-square tests. The prevalence of sunburn last summer was tested against family holidays abroad in the last year with a Chi-square test. Due to the skewed distribution in tanning bed use variables (only a small proportion of young people reported having ever used a tanning bed), bootstrapping was applied with 500 resampling for each statistical test. For Chi-square tests, the effect size Cramer's $V$ indices, alongside their bootstrapped 95% confidence intervals (95% CI) are presented. Statistical significance for all analyses was set at $p < 0.05$. For interpreting Cramer's $V$ indices, we followed the guidelines from Cohen [30], deeming values $\leq 0.10$ "small", $\leq 0.30$ "medium" and $\leq .50$ "large" effects.

Only a relatively small number of children reported ever using tanning beds but there were inconsistent responses on the four circumstance items. For instance, very few children (around 0.1–0.2%) reported never using a tanning bed but subsequently reported circumstances of use. A relatively larger proportion (around 4.8–5.0%) reported never using a tanning bed and subsequently did not answer the circumstances of tanning bed use, despite the first response option there was "Never used a tanning bed". Responses were cleaned to include all who responded positively to three or more of these five questions.

## RESULTS

Due to the large volume of these results, prevalence data on sun and UV-related behaviors in gender, age and social class breakdown are reported in the Supplementary Material (**Supplementary Tables S1–S39**).

## Sun Protection

More than half of the children reported never wearing a hat on a sunny day, and almost half reported never wearing clothes that cover their limbs. A large proportion reported sometimes wearing sunglasses and using sunscreen (**Table 1**). There were, however, 312 children (3.0%) who reported never using any form of sun protection.

Gender was associated with sun protection behaviors, but the effect sizes did not exceed medium (**Table 2**). Boys were more likely than girls to report always or sometimes using a hat, while girls were more likely than boys to never use a hat (**Supplementary Table S1**). Girls were also more likely than boys to wear sunglasses, clothes that protect their limbs, avoid sun between 12 and 3pm, and use sunscreen (**Supplementary Tables S4, S7, S10, S13**, respectively). All sun protection behaviors were significantly associated with age and social class, but the effect sizes were low (**Tables 3, 4**). Older children were more likely than younger children to always wear sunglasses, always or sometimes wear protective clothing, but less likely than younger children to wear a hat, avoid sun between 12 and 3pm, or always use sunscreen (**Supplementary Tables S2, S5, S8, S11, S14**, respectively). Children from lower social class groups were more likely to sometimes or never wear a hat, avoid the sun between 12 and 3 pm and never use sunscreen, but more likely to report always or sometimes wearing protective clothes than children from higher socio-economic groups (**Supplementary Tables S3, S6, S9, S12, S15**, respectively).

## Sunburn

Sunburn was experienced by most children (90% lifetime and 74% last summer). However, one in ten children reported sunburn five or more times last summer, and 44% reported five or more lifetime sunburn episodes (**Table 1**).

Gender, age, and social class were significantly associated with reported sunburn, but effect sizes were low (**Tables 2–4**). Boys were most likely to report no sunburn last summer (**Supplementary Table S16**), but there was no gender difference in lifetime sunburn (**Supplementary Table S19**). The frequency of sunburn episodes, both during last summer and lifetime, accumulated with age: younger children were more likely than older children to report no sunburn or experienced it once or twice, while older children were more likely than younger children to report five or more sunburn episodes (**Supplementary Tables S17, S20**). Children from lower social class groups were more likely than those from higher social class groups to report either none or five or more sunburn episodes last summer, while intermediary number of occasions (one to four) were more likely to be reported by children with higher social class groups than those from lower social class groups (**Supplementary Tables S18, S21**).

## Indoor Tanning Bed Use

The vast majority of children had never used an indoor tanning bed, either in the last 12 months or in their lifetime (95%), and most of those who did reported doing so once (**Table 1**). Using an indoor tanning bed was significantly associated with gender (**Table 2**) and social class (**Table 4**), but not age (**Table 3**). Girls and children from lower social class groups were most likely to report tanning bed use than

**TABLE 1 |** Frequency of Sun protection and UV- related behaviours in the in the 2018 Health Behaviour in School-aged Children study in Ireland (n = 10,271).

**Sun protection**

|  | Always | Sometimes | Never | Missing |
|---|---|---|---|---|
| Wearing a hat on a sunny day | 4.0% (410) | 42.2% (4,331) | 51.5% (5291) | 2.3% (239) |
| Wearing sunglasses on a sunny day | 14.6% (1496) | 56.8% (5836) | 26.9% (2762) | 1.7% (177) |
| Wearing clothes that cover arms and legs | 6.0% (620) | 42.9% (4,404) | 49.1% (5048) | 1.9% (199) |
| Avoiding the sun between 12 and 3pm | 3.3% (338) | 28.6% (2935) | 66.1% (6785) | 2.1% (213) |
| Using sunscreen | 30.7% (3150) | 51.1% (5248) | 17.3% (1781) | 0.9% (92) |

**Sunburn**

|  | Never | 1 time | 2 times | 3–4 times | 5 times or more | Missing |
|---|---|---|---|---|---|---|
| Getting sunburnt last summer | 26.0% (2666) | 25.3% (2594) | 21.9% (2249) | 15.1% (1553) | 9.7% (997) | 2.1% (212) |
| Getting sunburnt lifetime | 11.5% (1177) | 10.1% (1042) | 11.3% (1156) | 20.1% (2061) | 44.4% (4,556) | 2.7% (279) |

**Using an indoor tanning bed**

|  | Never | 1 time | 2 times | 3–4 times | 5 times or more | Missing |
|---|---|---|---|---|---|---|
| Using an indoor tanning bed last 12 months | 95.0% (9753) | 1.1% (108) | 0.5% (51) | 0.4% (36) | 0.7% (70) | 2.5% (253) |
| Using an indoor tanning bed lifetime | 94.9% (9747) | 1.1% (111) | 0.5% (49) | 0.6% (66) | 1.0% (104) | 1.9% (194) |

**Circumstances of using an indoor tanning bed [a]**

|  | Never used a tanning bed [b] | Yes, every time | Yes, at least once | No | Don't remember | Missing |
|---|---|---|---|---|---|---|
| Asked about age | – | 13.8% (40) | 23.9% (69) | 35.3% (102) | 23.9% (69) | 3.1% (9) |
| Being told to wear protective goggles | – | 30.1% (87) | 14.2% (41) | 37.0% (107) | 16.3% (47) | 2.4% (7) |
| Being given advice on skin type | – | 20.1% (58) | 19.0% (55) | 41.2% (119) | 19.0% (55) | 0.7% (2) |
| Told about health risks | – | 24.2% (70) | 16.6% (48) | 40.5% (117) | 17.6% (51) | 1.0% (3) |

[a]Cleaned for a combination of 3, 4, or 5 positive responses of using an indoor tanning bed lifetime and circumstances of using a tanning bed (n = 289).
[b]Those reporting 'Never used a tanning bed' on these items were excluded from the analysis.

**TABLE 2 |** Sun protection and UV- related behaviours across genders in the 2018 Health Behaviour in School-aged Children study in Ireland.

|  | $\chi^2$ | p | V | 95% CI |
|---|---|---|---|---|
| **Sun protection** | (df = 2) |  |  |  |
| Wearing a hat on a sunny day (n = 10,032) | 68.95 | < 0.001 | 0.083 | 0.065–0.104 |
| Wearing sunglasses on a sunny day (n = 10,094) | 824.45 | < 0.001 | 0.286 | 0.267–0.303 |
| Wearing clothes that cover arms and legs (n = 10,072) | 12.67 | 0.002 | 0.035 | 0.019–0.056 |
| Avoiding the sun between 12 and 3pm (n = 10,058) | 98.02 | < 0.001 | 0.099 | 0.080–0.117 |
| Using sunscreen (n = 10,179) | 360.28 | < 0.001 | 0.188 | 0.169–0.207 |
| **Sunburn** | (df = 4) |  |  |  |
| Getting sunburnt last summer (n = 10,059) | 44.41 | < 0.001 | 0.066 | 0.049–0.089 |
| Getting sunburnt lifetime (n = 9992) | 9.30 | 0.054 | 0.030 | 0.016–0.054 |
| **Using an indoor tanning bed** | (df = 4) |  |  |  |
| Using an indoor tanning bed last 12 months (n = 10,018) | 12.61 | 0.013 | 0.035 | 0.021–0.056 |
| Using an indoor tanning bed lifetime (n = 10,077) | 11.58 | 0.021 | 0.034 | 0.019–0.056 |
| **Circumstances of using an indoor tanning bed[a]** | (df = 3) |  |  |  |
| Asked about age (n = 280) | 2.84 | 0.417 | 0.101 | 0.040–0.238 |
| Being told to wear protective goggles (n = 282) | 12.72 | 0.005 | 0.212 | 0.121–0.349 |
| Being given advice on skin type (n = 287) | 24.23 | < 0.001 | 0.291 | 0.192–0.414 |
| Told about health risks (n = 286) | 21.29 | < 0.001 | 0.273 | 0.158–0.391 |

[a]Cleaned for a combination of 3, 4, or 5 positive responses of using an indoor tanning bed lifetime and circumstances of using a tanning bed.

boys and those from higher social class groups, though the effect sizes were low. Last 12 months and lifetime prevalence data on tanning bed use across genders are reported in **Supplementary Tables S22, S25**; across age groups in **S23** and **S26**; across social classes in **S24** and **S27**, respectively.

# Circumstances of Indoor Tanning Bed Use

Among those who responded that they had used indoor tanning beds, 14% reported always being asked about their age, 20% always received advice on their skin type, 24% were always told about health risks, and 30% were always instructed to wear

**TABLE 3 |** Sun protection and UV- related behaviours across age groups in the 2018 Health Behaviour in School-aged Children study in Ireland.

|  | $\chi^2$ | *p* | *V* | 95% CI |
|---|---|---|---|---|
| **Sun protection** | (*df* = 4) | | | |
| Wearing a hat on a sunny day (*n* = 10,032) | 259.40 | < 0.001 | 0.114 | 0.101–0.128 |
| Wearing sunglasses on a sunny day (*n* = 10,094) | 15.19 | 0.004 | 0.027 | 0.016–0.043 |
| Wearing clothes that cover arms and legs (*n* = 10,072) | 60.26 | < 0.001 | 0.055 | 0.043–0.069 |
| Avoiding the sun between 12 and 3pm (*n* = 10,058) | 54.83 | < 0.001 | 0.052 | 0.038–0.067 |
| Using sunscreen (*n* = 10,179) | 351.93 | < 0.001 | 0.131 | 0.119–0.147 |
| **Sunburn** | (*df* = 8) | | | |
| Getting sunburnt last summer (*n* = 10,059) | 110.80 | < 0.001 | 0.074 | 0.062–0.090 |
| Getting sunburnt lifetime (*n* = 9992) | 221.72 | < 0.001 | 0.105 | 0.093–0.121 |
| **Using an indoor tanning bed** | (*df* = 8) | | | |
| Using an indoor tanning bed last 12 months (*n* = 10,018) | 14.51 | 0.070 | 0.027 | 0.021–0.045 |
| Using an indoor tanning bed lifetime (*n* = 10,077) | 12.06 | 0.149 | 0.024 | 0.019–0.043 |
| **Circumstances of using an indoor tanning bed**[a] | (*df* = 6) | | | |
| Asked about age (*n* = 280) | 9.62 | 0.142 | 0.131 | 0.100–0.237 |
| Being told to wear protective goggles (*n* = 282) | 5.60 | 0.470 | 0.100 | 0.064–0.216 |
| Being given advice on skin type (*n* = 287) | 16.78 | 0.010 | 0.171 | 0.130–0.267 |
| Told about health risks (*n* = 286) | 13.99 | 0.030 | 0.156 | 0.114–0.254 |

[a]*Cleaned for a combination of 3, 4, or 5 positive responses of using an indoor tanning bed lifetime and circumstances of using a tanning bed.*

**TABLE 4 |** Sun protection and UV-related behaviours across social class groups in the 2018 Health Behaviour in School-aged Children study in Ireland.

|  | $\chi^2$ | *p* | *V* | 95% CI |
|---|---|---|---|---|
| **Sun protection** | (*df* = 4) | | | |
| Wearing a hat on a sunny day (*n* = 10,032) | 15.02 | 0.005 | 0.027 | 0.017–0.044 |
| Wearing sunglasses on a sunny day (*n* = 10,094) | 5.04 | 0.283 | 0.016 | 0.008–0.033 |
| Wearing clothes that cover arms and legs (*n* = 10,072) | 25.15 | < 0.001 | 0.035 | 0.024–0.053 |
| Avoiding the sun between 12 and 3pm (*n* = 10,058) | 13.81 | 0.008 | 0.026 | 0.016–0.043 |
| Using sunscreen (*n* = 10,179) | 54.33 | < 0.001 | 0.052 | 0.036–0.070 |
| **Sunburn** | (*df* = 8) | | | |
| Getting sunburnt last summer (*n* = 10,059) | 31.71 | < 0.001 | 0.040 | 0.030–0.058 |
| Getting sunburnt lifetime (*n* = 9992) | 44.55 | < 0.001 | 0.047 | 0.037–0.066 |
| **Using an indoor tanning bed** | (*df* = 8) | | | |
| Using an indoor tanning bed last 12 months (*n* = 10,018) | 31.44 | < 0.001 | 0.040 | 0.031–0.059 |
| Using an indoor tanning bed lifetime (*n* = 10,077) | 31.97 | < 0.001 | 0.040 | 0.031–0.059 |
| **Circumstances of using an indoor tanning bed**[a] | (*df* = 6) | | | |
| Asked about age (*n* = 280) | 7.38 | 0.287 | 0.115 | 0.090–0.223 |
| Being told to wear protective goggles (*n* = 282) | 8.33 | 0.215 | 0.122 | 0.088–0.238 |
| Being given advice on skin type (*n* = 287) | 3.87 | 0.694 | 0.082 | 0.067–0.187 |
| Told about health risks (*n* = 286) | 2.12 | 0.908 | 0.061 | 0.055–0.179 |

[a]*Cleaned for a combination of 3, 4, or 5 positive responses of using an indoor tanning bed lifetime and circumstances of using a tanning bed.*

protective goggles. Overall, 35–40% of users reported that the given safety measure was not advised (**Table 1**).

For some behaviors, significant associations were found with gender and age, but the effect sizes were low (**Tables 2–3**). Two notable exceptions are the association of gender with getting advice on skin type and being told about health risks, where medium effect sizes were detected. Girls were more likely than boys to report that these safety measures were never advised when they used an indoor tanning bed. The age difference was primarily attributed to younger children being more likely than older children to report that they had not encountered any of the safety measures when they used an indoor tanning bed. Circumstances of tanning bed use were not associated with social class (**Table 4**). Prevalence data on the circumstances of tanning

bed use across genders are reported in **Supplementary Tables S28, S31, S34, S37**; across age groups in **S29, S32, S35** and **S38**; across social class groups in **S30, S33, S36** and **S39**, respectively.

## Sunburn and Family Holidays Abroad

Reporting having had family holiday(s) abroad was linearly associated with frequency of sunburn: $\chi^2$ (12) = 24.96, *p* = 0.015, however the effect was low-sized: *V* = 0.029 [95% CI = 0.025–0.046]. Those who have not had any holidays abroad were the least likely to report ever being sunburnt; those who had one such holiday were most likely to report one episode of sunburn; those who had two holidays abroad were most likely to report two sunburns; and those who reported more

**TABLE 5 |** Getting sunburnt last summer and having had foreign holiday(s) in the last year in the 2018 Health Behaviour in School-aged Children study in Ireland (*n* = 9616).

| | | Getting sunburnt last summer | | | | |
| --- | --- | --- | --- | --- | --- | --- |
| | | **Never** | **1 time** | **2 times** | **3–4 times** | **5 times or more** |
| Foreign holidays last year | Not at all | 23.6% (604) | 22.4% (555) | 20.5% (436) | 22.1% (328) | 22.3% (209) |
| | Once | 39.4% (1005) | 41.9% (1039) | 40.3% (869) | 39.9% (592) | 37.1% (348) |
| | Twice | 19.2% (490) | 20.7% (513) | 22.2% (478) | 20.5% (305) | 21.5% (201) |
| | More than twice | 17.8% (455) | 15.1% (375) | 17.4% (375) | 17.5% (260) | 19.1% (179) |

*Note. Percentages are displayed for the proportion within the given sunburn frequency.*

than two holidays were most likely to report having five or more episodes of sunburn (**Table 5**).

# DISCUSSION

## Sun- and UV-Related Behaviors Among Schoolchildren in Ireland

This study found that while sunscreen application and wearing sunglasses on a sunny day were common, other sun safety behaviors were less consistently used. Only half of children reported wearing a hat or long-sleeved clothing in the sun. The older the child, the lower was the likelihood of sunscreen use. Girls more frequently reported using sunscreen, but they were also more likely to engage in the use of indoor tanning beds [11]. These baseline patterns are broadly aligned with those from other countries [6–8, 14, 15]. They can inform assessments of population skin cancer risk posed by the exposure of children in Ireland to UV radiation.

Last year sunburn estimates, at 74%, were similar to recent estimates from Northern Ireland (78%) [12]. Most adolescents in Ireland have Fitzpatrick skin type I or II (pale skin, blue/green eyes and light hair), indicating a significant population-level genetic predisposition to sunburn and future skin cancer [31]. Only 10% of the Irish sample reported that they had never been sunburnt. Sunburn was reported more frequently among children in high and middle socio-economic groups. This pattern was observed in both last summer and lifetime experience of sunburn. This may be related to a higher likelihood of higher and middle socio-economic groups travelling to sunny destinations during holidays abroad. Since we did not have an item specifying holiday destinations, this cannot be investigated. While our survey did not allow for differentiating between holiday destinations, we observed an association between the number of holidays and episodes of sunburn last summer. An analysis on climate preferences of Irish tourists demonstrated that families with children under 13 years preferred destinations with a high temperature [32]. This provides some supportive evidence to the hypothesis that family holidays abroad increase the risk of sunburn. Families taking holidays in sunny countries represent a distinct target group. Designing, delivering, and evaluating effective health behavior change interventions with a focus on "sun holiday risk" may be beneficial.

## Policy Implications

We found that over 90% of the children reported lifetime experience of sunburn (74% last summer). High levels of sunburn reported in childhood presents an ongoing risk for future skin cancer diagnosis at population level and a challenge for intervention [16].

Sunscreen was the primary method of sun protection among children in Ireland. There is a need to shift the pattern of sun safety behavior from an over-reliance on a single measure like sunscreen to more comprehensive practices employing multiple protection measures, such as wearing a hat, covering arms and legs, avoiding sun during peak UV hours, and wearing sunglasses. It is concerning that 3% of the children applied none of the protective measures. Building a habit of multi-measure sun safety behaviors in childhood can lead to increased protection in adulthood [33].

Reducing rates of sun overexposure or sunburn were key indicators for measuring progress in other jurisdictions such as in Australian national and regional policy. Successful and sustained implementation of the SunSmart Program contributed to an overall decline in melanoma rates from 1987 to 2017 [34, 35]. The publication of the recent Irish *National Skin Cancer Prevention Plan* (2019–2022) [5] will require monitoring of sunburn rates among children over time to assess the effectiveness of behavior change interventions for young people.

Drawing on a systematic evidence review on interventions to prevent excessive UV exposure [36], the United States Task Force on Community Preventive Services recommend community interventions with multiple components, including media campaigns directed at individuals and policy to improve environmental protections in the settings where both parents and children live, work/study and play [37]. Educational interventions in primary schools are also valid as an evidence-based intervention to increase children's covering-up behavior, specifically protective clothing and hats [36].

### Indoor Tanning Beds

A small but remarkable number of children (3%) reported using indoor tanning beds, a device that emits high levels of UV radiation. In July 2009, the WHO classified indoor tanning beds as a highly carcinogenic to humans. A secret shopper exercise found violations of legislation among 40 indoor tanning bed operators in the greater Dublin area. Despite the

prohibition, one third of the operators agreed to a booking with an underage user, even when the user failed to provide state issued identification to verify age [38]. The introduction of additional legislation and regulation banning tanning beds can help to address underage access and prevent skin cancer [39]. Other jurisdictions such as Brazil (2009) and Australia (2016) have passed complete bans on indoor tanning beds [40].

Girls and children from lower social class groups were most likely to report indoor tanning bed use than their more advantaged peers. Girls were more likely than boys to report that they had not been given advice on their skin type and were not told about health risks. Policy interventions and awareness measures will need to be carefully designed to target these groups. The marketing and promotion techniques for purchasing indoor tanning bed access also need to be monitored and studied to understand the ways in which industry influences tanning desirability among children. Although we found no Irish studies in this area, international evidence shows that the indoor tanning industry employs marketing strategies which downplay health risks, emphasize physical attractiveness and target specific subgroups, including young women [41].

## Strengths and Limitations

This is the first nationally representative study describing sunburn, sun safety and indoor tanning experiences among school children in Ireland. The data represent baseline estimates to inform implementation and monitoring of Ireland's first *National Skin Cancer Prevention Plan 2019–2022* [5]. These data were collected using a rigorous methodology and can pave the way for future cross-national comparisons if other countries collecting HBSC data follow suit. We hope that the items presented here are suitable for different countries and will therefore enable future comparative work.

However, this study has some limitations. We did not collect data on the severity of sunburn, whether children's hats had a wide brim, their attitudes and perceptions around tanning and UV exposure, or whether tanning bed use occurred in a tanning salon or another venue (e.g., in the family home). The distinction between sun safety behaviors when children are on holidays abroad versus in Ireland was not assessed. The sample is limited to those aged 10–17 years. Self- or parent-reported data from younger children are needed to develop a life-course perspective on predictors of future skin cancer rates. We have not assessed whether children applied sunscreen on their whole body or only on their face. No data was collected on the Sun Protection Factor (SPF) of the sunscreen used by the children. It is recommended that young people in Ireland use a sunscreen with SPF 50, in compliance with government recommendations [5]. However, it remains to be explored whether children (especially in younger age groups) understand the concept of SPF. Season and destination of family holidays abroad were not assessed, though families

travelling with children do report holiday preferences for hot destinations [32]. We recognize the limitations of only including three response options and the breadth of the option "sometimes" in the sun protection items.

Finally, care should be taken in the interpretation of the data on indoor tanning bed use due to the low sample sizes and the reported inconsistencies in responses to these questions. We cannot infer the cause of these, nevertheless our pilot study [20] noted a very low number of children who did not understand the concept of indoor tanning beds. It is possible that some children did not respond to the items on circumstances of tanning bed use because they perceived them as redundant.

## Conclusion

The findings of this study have important implications for public health practice, research, and monitoring. The documented gender age, and social class differences in risk behaviors can help shape the design and targeting of behavior change and regulatory interventions to reduce and prevent UV exposure in young people. There is a need to develop comparable datasets across Europe and monitor sun safety and tanning bed use trends over time. These will inform further policies and help develop targeted interventions combating modifiable risk factors for future skin cancer.

## DATA AVAILABILITY STATEMENT

The datasets analyzed in this study can be accessed in accordance with the HBSC data access policy: http://www.nuigalway.ie/hbsc/dataaccess/.

## ETHICS STATEMENT

This study was reviewed and approved by the Research Ethics Committee, National University of Ireland Galway. Informed consent to participate in this study was provided by the participants' legal guardian/next of kin.

## AUTHOR CONTRIBUTIONS

This paper is a product of collaboration between the Institute of Public Health in Ireland (LR and HMcA) and the HBSC Ireland team (AK and SNG). LR and HMcA wrote the first draft of Introduction and Discussion, and they secured funding for the article processing charge. AK and SNG carried out the data collection and the statistical analysis and wrote the first draft of Methods and Results. SNG secured funding for the survey and oversaw and supervised the data analysis and writing process. All authors took part in developing the survey items on sun protection and UV-related behaviors. All authors have critically revised the manuscript and approved its final version.

# FUNDING

The survey was funded by the Department of Health, Republic of Ireland.

# CONFLICT OF INTEREST

The authors declare that the research was conducted in the absence of any commercial or financial relationships that could be construed as a potential conflict of interest.

# ACKNOWLEDGMENTS

We thank all the children, parents and schools who facilitated data collection; Mary Scarlett (Public Health Information and Research Branch, Department of Health Northern Ireland), Ciara Reynolds (Institute of Public Health) and colleagues at the National Cancer Control Programme for their support; and our HBSC Ireland study colleagues who assisted with sampling, data collection and preparation of the dataset. HBSC is an international study carried out in collaboration with the WHO Regional Office for Europe. The international coordinator for the 2017/2018 study was Joanna Inchley, University of Glasgow. The Data Bank Manager was Oddrun Samdal, University of Bergen. In Ireland, the study has been carried out since 1998 by HBSC Ireland at the National University of Ireland Galway, commissioned by the Department of Health (Republic of Ireland). For details of the international study, see http://www.hbsc.org. For details of HBSC Ireland, see http://www.nuigalway.ie/hbsc/.

# SUPPLEMENTARY MATERIAL

The Supplementary Material for this article can be found online at: https://www.ssph-journal.org/articles/10.3389/ijph.2021.1604045/full#supplementary-material

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
