## [Reviewer comments · International Journal of Public Health]

Peer Review Report

Review Report on Sunburn, sun safety and indoor tanning among schoolchildren in Ireland

Original Article, Int J Public Health

Reviewer: Lesley Roberts

Submitted on: 25 Mar 2021

Article DOI: 10.3389/ijph.2021.1604045

EVALUATION

Q 1 Please provide your detailed review report to the authors. The editors prefer to receive your review structured in major and minor comments. Please consider in your review the methods (statistical methods valid and correctly applied (e.g. sample size, choice of test), is the study replicable based on the method description?), results, data interpretation and references. If there are any objective errors, or if the conclusions are not supported, you should detail your concerns.

This is a well presented and robustly conducted study which provides novel and important data on sun protection behaviours in school children in Ireland. The large sample size supports the value of findings to inform PH strategies in this arena in Ireland and analyses are appropriate with clear explanation as to how data discrepancies have been managed per-analysis. Conclusions flow directly from results presented.

Major comments:

1. The procedure for data collection is largely defined through reference to the protocol of the international HBSC network. As the referenced protocol requires registration access it is not immediately accessible by readers wishing to understand more about procedure applied. Furthermore the protocol explains that participants were notified of the 'confidential nature of answers' and also identifies that the survey is deliverable by teachers / researchers / school nurses etc. These two points perhaps need inclusion in the main text as they have direct implications for influence and bias in responses. If possible it would be helpful to also provide a little more detail on these items in relation to the procedure undertaken in this study and authors should consider whether this requires further comment in limitations section.
2. Tables 3 -5 are not accessible as stand-alone information sources. For example table 3 reports on behaviours by gender but does not present proportions by gender. Cross reference to the text informs that for example boys are more likely to wear a hat and girls more likely to wear sunglasses but as a table this reverse observation is not evident. The same applies to age group and social class tables 4 and 5 - whilst appreciating that inclusion of proportions within each group would lengthen tables significantly this data does seem key to facilitating interpretation and I have concerns that its absence may lead to mis-interpretation.

Minor comments:

None

Q 2 Please summarize the main findings of the study.

The main findings of the study include sunburn frequency (life-time and last year) for school children in Ireland - at 90% and 74% respectively this presents an important finding. Uptake of various sun protection strategies are presented and highlight low levels of adoption of sun protection behaviours such as midday sun avoidance (never adopted by 66%) and wearing a hat (never adopted by 52%) giving clear steer as to behaviours which should be targeted. Differences in behaviours by gender, age and social class give further detail which will advise targeted intervention, albeit that effect sizes are small in most cases. Association between sunburn frequency and family holidays abroad was identified and further indicates focus for future intervention.

The study also reports a low level of sunbed use, but recognition of the presence of this use is important. Data relating to risk advice and safety procedures in sunbed use are more difficult to interpret due to the small sample in this group.

Q 3 Please highlight the limitations and strengths.

The study has significant strengths in its use of data collected through the HBSC Ireland study, which as part of the WHO collaborative study with single aligned protocol enables cross nation comparisons. The large sample and robust survey strategy with required piloting of local variations add to the strengths of this work. Limitations are few but as a stand alone paper a little more information on procedures which may have impacted responder biases is necessary and not currently explored. Sub-group analyses of between group experiences of sunbed use are limited by the sample size in this sub-group but this is inevitable and is not over-played by the authors in presentation.

PLEASE COMMENT

Q 4 Is the title appropriate, concise, attractive?

Yes - this is highly descriptive title and the study does what one would expect from the title!

Q 5 Are the keywords appropriate?

Yes

Q 6 Is the English language of sufficient quality?

Yes - exceptionally well written and appropriately structured

Q 7 Is the quality of the figures and tables satisfactory?

Yes.

Q 8 Does the reference list cover the relevant literature adequately and in an unbiased manner?)

Yes - the authors have referenced appropriate literature and appear to have taken into account quality of those studies they have identified as relevant, clearly favouring larger studies with fewer risks of bias.

QUALITY ASSESSMENT

Q 9 Originality

Q 10 Rigor

Q 11 Significance to the field

Q 12 Interest to a general audience

Q 13 Quality of the writing

Q 14 Overall scientific quality of the study

REVISION LEVEL

Q 15 Please take a decision based on your comments:

Minor revisions.